# Impact of the COVID-19 Pandemic on the Outcomes of Transarterial Chemoembolization in Patients with Hepatocellular Carcinoma: A Single Center Experience from a Developing Country

**DOI:** 10.3390/medicina58121701

**Published:** 2022-11-22

**Authors:** Aleksandar Filipović, Dragan Mašulović, Marko Živanović, Tamara Filipović, Dušan Bulatović, Miloš Zakošek, Dejan Nikolić, Danijel Galun

**Affiliations:** 1Faculty of Medicine, University of Belgrade, 11000 Belgrade, Serbia; 2Center for Radiology, University Clinical Centre of Serbia, 11000 Belgrade, Serbia; 3HPB Unit, Clinic for Digestive Surgery, University Clinical Centre of Serbia, 11000 Belgrade, Serbia; 4Institute for Rehabilitation, 11000 Belgrade, Serbia; 5Department of Physical Medicine and Rehabilitation, University Children’s Hospital, 11000 Belgrade, Serbia

**Keywords:** COVID-19, transarterial chemoembolization, loco-regional therapy, survival, hepatocellular carcinoma

## Abstract

*Background and Objectives*: Treatment of cancer patients during the COVID-19 pandemic has been a challenge worldwide. In accordance with the current recommendations for hepatocellular carcinoma (HCC) management during the COVID-19 pandemic, loco-regional therapy such as transarterial chemoembolization (TACE) was proposed with the purpose of achieving local tumor control and improving overall survival. The aim of this prospective cohort study was to evaluate the outcomes of TACE treatment in patients with HCC during the COVID-19 pandemic in comparison with the outcomes of patients treated in the pre-pandemic period. *Materials and Methods*: Between September 2018 and December 2021, 154 patients were managed by serial TACE procedures for different liver tumors. Ninety-seven patients met the study criteria and were divided into two groups: the study group *n* = 49 (patients treated from May 2020 to December 2021); the control group *n* = 48 (patients treated from September 2018 to May 2020). *Results*: The mean waiting time for TACE was significantly longer in the study group compared to the control group (*p* < 0.001). No significant difference in survival between the groups is noted (log-rank test *p* = 0.823). In multivariate analysis, the MELD score (HR 1.329, 95% CI 1.140–1.548, *p* < 0.001) remained a significant predictor of mortality. *Conclusions*: COVID-19 pandemic did not affect the final outcome of TACE treatment.

## 1. Introduction

The pandemic has compromised elective interventions globally and created a risk of an impending collapse of the healthcare system due to a surge of patients with COVID-19. A certain pandemic “effect of distraction” on the continuous care of cancer patients has been registered [1]. Even in developed countries, different healthcare institutions and cancer centers have reoriented their activities to accommodate a large number of patients with COVID-19 disease [2]. An evident shortage of healthcare personnel and inpatient beds has dramatically limited opportunities for non-COVID procedures. During the COVID-19 pandemic, only emergency cases were hospitalized, while the treatment of chronic pathologies was delayed [3]. Delaying or canceling so many interventions can have a devastating impact on the medical system, patient prognosis, and patient quality of life [4]. The characteristics of the procedures performed changed, as did the addressability, and the overall structure of the activity shifted toward supporting the emergency system and prioritizing acute strategies vs. elective surgery [4]. The consequences of this situation were mainly reflected in cancer care worldwide [5]. Treatment of cancer patients during the COVID-19 pandemic has been a worldwide challenge; in particular, developing countries were at risk of facing overwhelmed health facilities [6]. Clinical care during the pandemic required priority use of resources for patients with a greater chance of survival, especially in developing countries [7].

In this complex setting, hepatocellular carcinoma (HCC) treatment had to be performed, taking into account the risk of delaying potentially curative treatment and the evident risk of COVID-19 infection. Providing cancer treatment while minimizing the risk of exposure to the SARS-CoV-2 virus had to be carefully weighed.

In this situation, the nonsurgical therapies gained major attention with the goal of achieving tumor control and improving overall survival [8]. During an early phase of the COVID-19 pandemic, the consensus was reached that HCC patients should continue to receive loco-regional therapy with the choice of therapy discussed by the multidisciplinary board [9]. The advantages of loco-regional therapies are the following: (a) they are less invasive and require a shorter hospital stay than surgery, (b) they can be used as a “bridge therapy” for HCC patients who are candidates for curative-intent treatment options, (c) the patients are exposed to reduced risk for acquiring COVID-19 infection in health care facilities and (d) in some cases loco-regional therapies are performed as a definite treatment [10].

Among different loco-regional treatment options, transarterial chemoembolization (TACE) was the most frequently used treatment modality in HCC patients. However, there is scarce data about the outcomes of the TACE procedure for HCC patients performed during the COVID-19 pandemic. While limited data are available from developed countries, there are no reports analyzing the outcomes of TACE treatment performed in developing countries. This is important, knowing that there is the disparate quality of healthcare between these countries reflected in screening and surveillance programs, available treatment modalities and drugs, and reimbursement policies.

The aim of this study was to evaluate the outcomes of TACE treatment in patients with HCC performed in a developing country during the COVID-19 pandemic.

## 2. Materials and Methods

### 2.1. Patients

Each patient was carefully evaluated by a multidisciplinary team in order to define the proper treatment strategy due to the COVID-19 infection risk and cancer treatment. All patients included in the study have presented a negative result of PCR test performed within 72 h prior to the TACE procedure.

Inclusion criteria for this study were: patients aged 18–75 years, pathological confirmation, or noninvasive criteria according to European Association for the Study of the Liver, Eastern Cooperative Oncology Group performance status 0–1, liver function Child-Pugh class A or B cirrhosis, no extrahepatic metastases, and TACE as the sole first-line anticancer treatment for HCC [11].

The exclusion criteria were: Eastern Cooperative Oncology Group performance status >2, previous curative anticancer treatment (liver resection or tumor ablation), extrahepatic dissemination of HCC, portal vein tumor thrombi with complete main portal vein occlusion and without adequate collateral circulation around the occluded portal vein, esophageal or gastric variceal bleeding or hepatic encephalopathy, severe cardiac and/or renal diseases, severe coagulation disorders, laboratory data and clinical signs of decompensated liver function (including ascites), impaired liver function (Child-Pugh score C).

Between September 2018 and December 2021, 154 patients were managed by serial TACE procedures for different liver tumors. Data were retrospectively reviewed from the prospectively maintained database of patients treated by TACE at the Clinic for Digestive Surgery, University Clinical Centre of Serbia, Department for hepato–bilio–pancreatic surgery. However, 57 patients were excluded from the current study because 33 obtained TACE for non-HCC indications, 19 received other curative-intent modalities as initial anticancer treatment (percutaneous ablation in 8, liver resection in 11), 3 had incomplete laboratory data, and 2 were lost to follow-up, resulting in a final enrollment of 97 patients (Figure 1).

The study endpoint was overall survival in days from the date of the TACE treatment.

### 2.2. Clinical and Laboratory Data Recording

Clinical and laboratory data were collected from all patients within 7 days prior to TACE. Clinical data included: patient age, sex, history of previous interventions and Child-Pugh score. In addition, imaging data were analyzed based on MDCT or MRI imaging. Laboratory data obtained the presence of hepatitis B and C infection, α-fetoprotein values, albumin (ALB) values, alanine aminotransferase (ALT), aspartate aminotransferase (AST), prothrombin time, platelet count (PLT) and total bilirubin (TBil). MELD score (Model for end-stage liver disease) was calculated by including total bilirubin, creatinine, and INR [12].

### 2.3. TACE Procedure

A standardized conventional TACE treatment was performed by two experienced interventional radiologists (with >10 years of experience).

Preprocedural multiphase MDCT examinations were used to analyze hepatic vascular anatomy, detect tumor feeders, select the appropriate leading angiographic catheter, and plan the superselective catheterization. The superselective embolization was performed by using a 2.6 Fr microcatheter (Asahi Intecc Co., Seto, Japan). The mixture of lipiodol (Lipiodol Ultra-Fluide, Guerbet, Roissy, France) and Cisplatin (PTY. Limited, Perth, Australia, Pfizer) was injected through microcatheter under fluoroscopy control until stasis was achieved in the second- or third-order branches of the lobar hepatic artery [13].

### 2.4. Follow-Up

The follow-up MDCT examinations for evaluation of the effects of therapy and decision-making on the second session were performed 4–8 weeks after the initial procedure. The assessment of response to therapy was performed using the modified RECIST criteria [14]. If intrahepatic recurrences were detected on a follow-up MDCT scan, the patient was evaluated for the feasibility of repeated TACE treatment. The sample size was not calculated in this pilot study because this is the first study to report the impact of the COVID-19 pandemic on the outcomes of transarterial chemoembolization in patients with hepatocellular carcinoma. However, post-hoc power testing indicated a 100% power of the study analyzing waiting time for TACE in two groups. The *t*-test for the independent means option was chosen from the test family as the statistical test in G*Power V.3.1.9.2. (University of Kiel, Kiel, Germany).

### 2.5. Statistical Analysis

Data are presented as arithmetic mean and standard deviation. The comparison of continuous variables to the fatal outcome was performed using the *t*-test or the Mann–Whitney test. The comparison of categorical variables was performed using the Chi-square test. Kaplan-Meier curve and Cox regression analysis were used to assess survival of COVID infection. The hypothesis was tested with the significance threshold value of *p* < 0.05. Statistical data analysis was performed in the SPSS 20.0 software package.

## 3. Results

Patients included in the study were divided into two groups: Group 1, the study group, were patients treated from May 2020 to December 2021 (pandemic period, *n* = 49); Group 2, the control group, were the patients treated in the pre-pandemic period from September 2018 to May 2020 (*n* = 48).

### 3.1. Clinical Characteristics

The median age at the time of the TACE procedure was 66 years (range 40–86). The majority of patients were male (67%). A total of 84 patients (86.6%) had a cirrhotic liver with preserved liver function (Child-Pugh A). The majority of patients had multiple tumors (60.8%). A major tumor size (≥10 cm) was present in 27.7% of patients. No statistically significant differences in demographic characteristics between the groups are noted. Clinical-pathologic and laboratory characteristics are also non-significantly different between groups (Table 1). The mean waiting time for TACE (in days) was significantly greater in Group 1 compared to Group 2 (*p* < 0.001).

### 3.2. Survival Analysis

The median follow-up was 14 months (1–35 months). At the median follow-up time, 65% of patients were alive. The median survivals were estimated as follows: Group 1—22.5 months (SE 1.8 months), Group 2—16.7 months (SE 1.1). The Kaplan-Meier curve demonstrated no significant difference in survival between the groups (log-rank test *p* = 0.823) (Figure 2).

### 3.3. Predictors of Survival

Univariate Cox regression demonstrated that a significant predictor is MELD (HR 1.294, 95% CI 1.125–1.487, *p* < 0.001). In multivariate analysis, the MELD score (HR 1.329, 95% CI 1.140–1.548, *p* < 0.001) remained a significant predictor of mortality (Table 2).

## 4. Discussion

The COVID-19 pandemic is expected to have a long-lasting impact on the approach to care for patients with HCC due to the risks from potential exposure and resource reallocation [15]. Operative treatment during the pandemic could be postponed due to the fact that the immunosuppressive effects of surgical stress could contribute to a higher risk of developing COVID-19 for patients with cancer [16,17]. Frequent postoperative hospital visits [18] and underlying liver disease can alternate both innate and adaptive immune responses [19].

Reduced access to the operating room could force the use of alternative treatment strategies outside the standard of care to delay surgery [20]. Transarterial chemoembolization is a cytoreductive procedure and could be performed with the goal of achieving local control and improving overall survival [20]. TACE could be performed as a “bridge treatment” for local disease control, mainly for patients with solitary tumors 3–5 cm in size or for multifocal HCC, and even for BCLC-C if the patient has portal vein thrombosis and no extrahepatic disease [21]. Patients with large tumor sizes (>5 cm), multifocal disease and vascular invasion could be treated with loco-regional transarterial therapy as a first-line treatment with the purpose of local disease control while waiting for surgery [22]. In the group of patients treated during the pandemic (*n* = 49), the majority of patients had multiple tumors (61.2%), and 27.7% of patients had a major tumor size ≥10 cm. For patients with unresectable HCC treated with cTACE, the reported rate of 5-year overall survival is 30% [23]. From this point in time, we are not able to compare our cTACE results with the previous statement since the median follow-up time in our series is 14 months. During this period, 65% of patients are alive.

During the COVID-19 pandemic, traditional indications for performing transarterial therapies are maintained [11,24,25], but certain updates for HCC management have been proposed by the European Association for the Study of the Liver (EASL) [10], the American Association for the Study of Liver Diseases (AASLD) [26] and the Asian Pacific Association for the Study of the Liver (APASL) [18].

For the group of oncology patients where the benefits of treatment outweigh the possible risk of SARS-CoV-2 infection, the priority might be given [27]. Thus multidisciplinary evaluation is recommended for all patients in order to identify those who would benefit the most from IR therapy [28]. The need for TACE should be estimated by assessing indications as the up-to-seven criteria [29,30] and albumin-bilirubin grade [31]. Therefore, during the pandemic, each patient was carefully evaluated by a multidisciplinary team in order to define the proper treatment strategy due to the COVID-19 infection risk and cancer treatment. A rigorous assessment of the individual risk-benefit ratio of TACE was estimated in light of the current pandemic status and insufficient resources for all to be treated. The follow-up imaging with the purpose of assessing the effects of therapy and decision-making on the second session was performed 4–8 weeks after the initial procedure. If intrahepatic recurrences were detected during follow-up, the patient was reevaluated through a multidisciplinary tumor board for the feasibility of repeated TACE treatment.

When compared to the general population, the risk of COVID-19 infection for cancer patients is increased by 3 times, the risk of severe infection is increased by 5 times, and the risk of death is increased by 8 times [32]. Therefore meticulous hygiene measures were taken in the hospital to reduce the risk of COVID-19 infection.

In patients with severe COVID-19 infection, a significant deterioration of liver function was shown [33,34]. This might be a result of cholangiocyte and hepatocyte injury [35,36,37], as well as immune-mediated liver injury and hypoxemia [35,36]. A recently published paper stated the possibility of drug-induced injury as the cause of liver dysfunction in patients with COVID-19 [38]. In HCC patients with a proven COVID-19 infection, transarterial chemoembolization should be postponed until recovery [2]. Thus it was mandatory for all patients to present a negative result of the PCR test performed within 72 h prior to the TACE procedure.

Another goal to achieve while performing TACE for HCC patients during the COVID-19 pandemic, bearing in mind the risk of infection, is to shorten the patients’ stay in the hospital. It has been shown that dexamethasone effectively reduces the occurrence of the post-embolization syndrome [39]. Thus short (3 days) dexamethasone therapy could be administered in TACE patients except those with contraindications in order to lower post-TACE symptoms and shorten the hospital stay [2]. No warning against short-term corticosteroid use in the beforementioned context is noted [40]. By using selective or super-selective chemoembolization, severe postembolization symptoms could also be prevented [18]. In all patients included in the study, a superselective approach was used, and the chemoembolization material was applied to the arterial feeder of the tumor through a microcatheter.

TACE should be suspended for patients with liver decompensation or comorbidities that increase the risk of severe COVID-19 [18,21]. Patients with Eastern Cooperative Oncology Group (ECOG) performance status ˃2, hepatic decompensation (including ascites) and impaired liver function (Child-Pugh score C) were not assigned to TACE treatment.

Due to the limited resources during the pandemic, a treatment delay is also noted. The mean waiting time for TACE (in days) was significantly greater in Group 1 compared to Group 2 (*p* < 0.001). Results from the published report of Amaddeo et al. indicate the possibility of HCC recurrence in patients with a treatment delay of more than 1 month [41].

Diagnostic and treatment delays for patients with hepatocellular carcinoma undergoing loco-regional therapy are defined as >90 days between HCC diagnosis and treatment [42,43,44]. For all patients included in the study, the time from HCC diagnosis to TACE treatment was 28.39 ± 8.39 days. The threshold value of 90 days was not reached.

Patients with a follow-up interval longer than 95 days might have a worse prognosis [45]. During this study, there were no timely intervals of control examinations which could have influenced a lower overall response rate for HCC. Depending on the availability of MDCT, the follow-up interval for both groups was in the range of 4–8 weeks.

The length of interval for the repeated TACE treatment is variable in the literature, with the range from 4–12 weeks [46,47], and still controversial [45]. Results from the published study by Yang and colleagues have shown no significant difference in the case of a long interval (>48 days) than a short interval (<48 days) between the first two TACE treatments for patients at BCLC stage A or B [48]. Kim et al. noted no significant difference in survival between patients undergoing repeated TACE with an interval shorter than 60 days in comparison with those having >60 days interval [49].

Monitoring of patients after HCC treatment ought to be performed with the principle of maximizing the risk-benefit ratio [50]. Patients with multinodular disease who did not have an adequate response after a second TACE, thus having a reduced chance of achieving an intended therapeutic effect, should be considered for systemic treatment [21].

## 5. Conclusions

The pandemic did not affect the final outcome of TACE treatment. There was no statistically significant difference in survival in days from TACE treatment between patients treated in the pre-pandemic period and patients treated during the COVID-19 pandemic. MELD score is a predictive factor in the mortality of patients with HCC. Allocating IR resources to patients with the highest chance of benefit is the rational approach in the current setting of limited healthcare delivery.

## Figures and Tables

**Figure 1 medicina-58-01701-f001:**
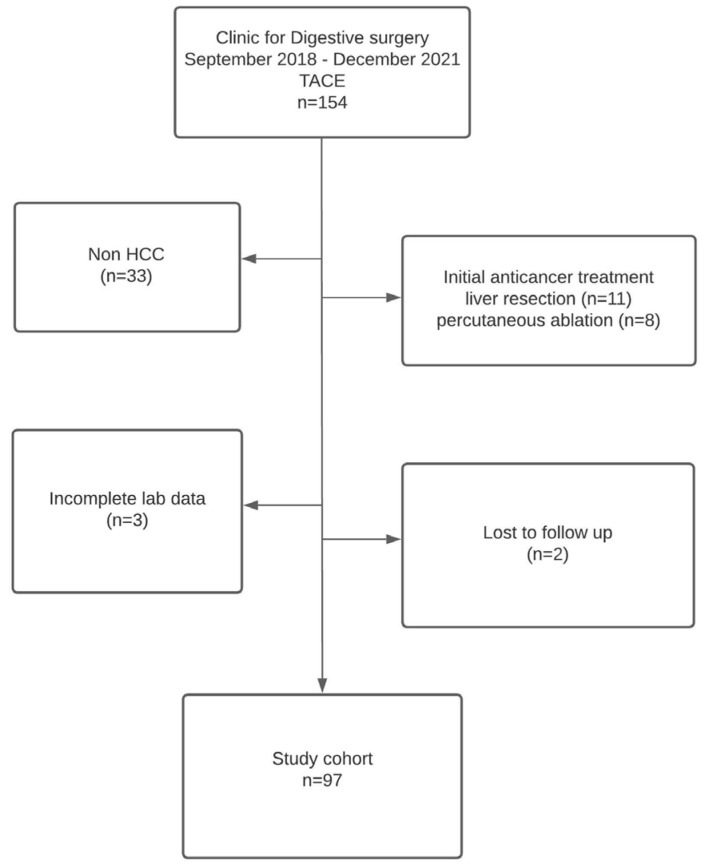
Patients Flowchart.

**Figure 2 medicina-58-01701-f002:**
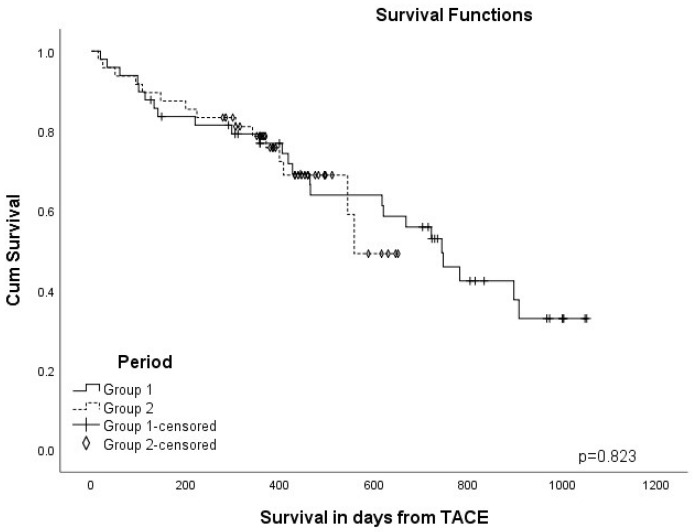
Overall survival of HCC patients treated by TACE.

**Table 1 medicina-58-01701-t001:** Demographic and clinical data.

Characteristic	Total		Group 1 (*n* = 49)		Group 2 (*n* = 48)		*p* ^1^
Age	66.67 ± 8.67		65.94 ± 8.68		67.42 ± 8.68		0.404
Sex							
Male	65	67.0	32	65.3	33	68.8	0.718 ^2^
Female	32	33.0	17	34.7	15	31.3	
COVID-19 infection	38	39.2	15	30.6	23	47.9	0.081 ^2^
Number of tumors							
Solitary	38	39.2	19	38.8	19	39.6	1.000 ^2^
Multiple	59	60.8	30	61.2	29	60.4	
Tumor diameter	81.93 ± 35.90		75.69 ± 30.78		88.29 ± 39.79		0.124 ^3^
BCLC staging							
B	92	94.8	47	95.9	45	93.8	0.981 ^2^
C	5	5.2	2	4.1	3	6.3	
Child status							
A	84	86.6	43	87.8	41	85.4	0.968 ^2^
B	13	13.4	6	12.2	7	14.6	
MELD	8.23 ± 1.88		8.12 ± 1.73		8.3 ± 2.05		0.717 ^3^
Waiting time for TACE (Days)	28.39 ± 8.39		22.59 ± 5.09		34.31 ± 6.82		<0.001
Bilirubin	23.95 ± 17.34		28.26 ± 19.32		22.34 ± 16.47		0.131 ^3^
AST	93.45 ± 128.86		74.78 ± 86.02		100.46 ± 141.79		0.368 ^3^
ALT	63.98 ± 73.52		54.17 ± 58.54		67.67 ± 78.64		0.485 ^3^

^1^*t*-test, ^2^ Chi-squared tests, ^3^ Mann–Whitney tests.

**Table 2 medicina-58-01701-t002:** Association of outcome and demographic and clinical characteristics. (univariate and multivariate Cox regression analysis).

Characteristic	Univariate				Multivariate			
	HR	HR 95% CI		*p*	HR	HR 95% CI		*p*
		Lower	Upper			Lower	Upper	
Sex	0.879	0.445	1.735	0.710	0.737	0.334	1.628	0.451
Age	1.036	0.995	1.078	0.087	1.032	0.989	1.077	0.145
COVID-19 infection	0.642	0.331	1.247	0.191	0.667	0.329	1.351	0.261
Diameter max (mm)	1.000	0.990	1.009	0.949	0.998	0.988	1.008	0.696
MELD	1.294	1.125	1.487	<0.001	1.329	1.140	1.548	<0.001

Hosmer–Lemeshow test *p* = 0.284.

## Data Availability

During the review process, links to publicly archived datasets will be provided.

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
