# Peer review of "Impact of the COVID-19 Pandemic on the Outcomes of Transarterial Chemoembolization in Patients with Hepatocellular Carcinoma: A Single Center Experience from a Developing Country"

_medicina, 2022, doi:10.3390/medicina58121701_

Round 1

Reviewer 1 Report

Dear Editor,

I have read the paper by Filipović et al with interest. The authors investigate the outcome of TACE treatment in patients with HCC during COVID-19 pandemic.

The Study found that the COVID-19 pandemic did not affect the outcome of TACE treatment. The paper is of novelty and present interest for the community. However, before publication there are some concerns that must be resolved.

Introduction:

1.I suggest in the Introduction section to add more information regarding the negative impact of COVID-19 pandemic over the medical activity. See and add the following articles: https://doi.org/10.3390/jcm11092452 , https://doi.org/10.3389/fsurg.2022.883935

Results:

2.Regarding the COVID-19 infection, as I seen in table 1, the pre-pandemic group (Group 2) have 23 patients with COVID-19 infection. I am wondering if the patients develop infection during the follow up. In this case will be interesting the compare the patients by dividing them in 2 groups depending on the COVID-19 infection.

3. Moreover, it will be interesting to compare the number of patients diagnosed with HCC, tumor diameter, BCLC staging, Child status, and waiting time for TACE for every patient admitted in the same period pre-pandemic vs pandemic (for example May 2018-December 2019 vs May 2020-December 2021). Of course, if the authors have such data. If not, maybe you can have this subject in plan for a future paper.

I want to congratulate the authors for their work.

Kind regards

Author Response

Introduction

Comment no 1: I suggest in the Introduction section to add more information regarding the negative impact of COVID-19 pandemic over the medical activity. See and add the following articles: https://doi.org/10.3390/jcm11092452 , https://doi.org/10.3389/fsurg.2022.883935

Reply: Based on the reviewer’s input the additional information regarding the negative impact of COVID-19 pandemic over the medical activity has been added in the Introduction section.

Results:

Comment no 2. Regarding the COVID-19 infection, as I seen in table 1, the pre-pandemic group (Group 2) have 23 patients with COVID-19 infection. I am wondering if the patients develop infection during the follow up. In this case will be interesting the compare the patients by dividing them in 2 groups depending on the COVID-19 infection.

Reply: We thank the reviewer for this comment. However, based on the results of univariate and multivariate analysis the MELD score was found the only predictor of survival. That is the reason why we did not perform additional analysis comparing the patient cohorts with and without Covid infection. Even in univariate analysis Covid infection was not found as a predictor of survival.

  1. Moreover, it will be interesting to compare the number of patients diagnosed with HCC, tumor diameter, BCLC staging, Child status, and waiting time for TACE for every patient admitted in the same period pre-pandemic vs pandemic (for example May 2018-December 2019 vs May 2020-December 2021). Of course, if the authors have such data. If not, maybe you can have this subject in plan for a future paper.

Reply: We fully agree with the reviewer that the propensity score analysis should follow the study that is presented in this paper. We intend to complete this analysis in the near future and to include more patients from longer observation period. 

Please see the attachment (revised version of the Manuscript).

Reviewer 2 Report

1. There are some typos in the manuscript. For example, "distraction effect" has...; Covid-19; ,,bridging. Please proof-read your manuscript carefully. 

2. In your exclusion criteria, you have performance status > 2. What does it mean? Please clarify. 

3. The authors mentioned no evidence could be referenced for statistical power analysis. However, if you know your statistical model, you could perform power analysis. For example, you used t-test for your data analysis. Then, you could do power analysis based on t-test. 

4. Cox regression analysis was used to assess survival of COVID infection and vaccination status. This is not correct. You did not consider any vaccination variable. 

5. You did not define MELD. 

6. Why did you use Mann-Whitney test? If the normal assumptions are not satisfied, you could use Mann-Whitney for all tests. 

Author Response

1.There are some typos in the manuscript. For example, "distraction effect" has...; Covid-19; ,,bridging. Please proof-read your manuscript carefully.

Reply: The necessary corrections are completed accordingly.

  1. In your exclusion criteria, you have performance status > 2. What does it mean? Please clarify.

Reply: The correction is made in the manuscript. It has been clarified that the exclusion criteria were Eastern Cooperative Oncology Group performance status >2.

  1. The authors mentioned no evidence could be referenced for statistical power analysis. However, if you know your statistical model, you could perform power analysis. For example, you used t-test for your data analysis. Then, you could do power analysis based on t-test.

Reply: The post hoc power analysis is done and incorporated into the manuscript in the follow-up section. Post hoc power analysis ensures that the study population was large enough and thus fit for the purpose.

  1. Cox regression analysis was used to assess survival of COVID infection and vaccination status. This is not correct. You did not consider any vaccination variable.

Reply: The vaccination status has been deleted from the sentence. We only assessed the survival of COVID-19 Infection.

  1. You did not define MELD.

Reply: According to your suggestion, the MELD score is defined in the Methods section/ Clinical and Laboratory Data Recording.

  1. Why did you use Mann-Whitney test? If the normal assumptions are not satisfied, you could use Mann-Whitney for all tests.

Reply: The Mann-Whitney test was used only for non-normally distributed variables.

Please see the attachment (revised version of the Manuscript)

Round 2

Reviewer 1 Report

no further comments

Reviewer 2 Report

The authors addressed all my concerns.